# Recent Advances in Light-Conversion Phosphors for Plant Growth and Strategies for the Modulation of Photoluminescence Properties

**DOI:** 10.3390/nano13111715

**Published:** 2023-05-23

**Authors:** Chengxiang Yang, Wei Liu, Qi You, Xiuxian Zhao, Shanshan Liu, Liang Xue, Junhua Sun, Xuchuan Jiang

**Affiliations:** 1Institute for Smart Materials & Engineering, School of Materials Science and Engineering, University of Jinan, No. 336 Nanxinzhuang West Road, Jinan 250022, China; 202211100036@stu.ujn.edu.cn (C.Y.); ism_youq@ujn.edu.cn (Q.Y.); mse_zhaoxx@ujn.edu.cn (X.Z.); ism_liuss@ujn.edu.cn (S.L.); ism_xuel@ujn.edu.cn (L.X.); 2School of Chemistry and Chemical Engineering, University of Jinan, No. 336 Nanxinzhuang West Road, Jinan 250022, China; sunjh102@hotmail.com

**Keywords:** plant growth, phosphors, light-conversion films, plant-growth LED, luminescent modulation

## Abstract

The advent of greenhouses greatly promoted the development of modern agriculture, which freed plants from regional and seasonal constraints. In plant growth, light plays a key role in plant photosynthesis. The photosynthesis of plants can selectively absorb light, and different light wavelengths result in different plant growth reactions. Currently, light-conversion films and plant-growth LEDs have become two effective ways to improve the efficiency of plant photosynthesis, among which phosphors are the most critical materials. This review begins with a brief introduction of the effects of light on plant growth and the various techniques for promoting plant growth. Next, we review the up-to-date development of phosphors for plant growth and discussed the luminescence centers commonly used in blue, red and far-red phosphors, as well as their photophysical properties. Then, we summarize the advantages of red and blue composite phosphors and their designing strategies. Finally, we describe several strategies for regulating the spectral position of phosphors, broadening the emission spectrum, and improving quantum efficiency and thermal stability. This review may offer a good reference for researchers improving phosphors to become more suitable for plant growth.

## 1. Introduction

With the booming development of modern agriculture, the techniques used to regulate plant growth have been increasingly well-established. Light, as the key source of all lives, plays an irreplaceable role in plant growth. However, it has been found that the plant growth is selective in light absorption, mainly due to the differences in light absorption by four main pigments in plants (chlorophyll A, chlorophyll B, photosensitive pigment P_R_ and photosensitive pigment P_FR_). Blue light (400–500 nm), red light (600–700 nm) and far-red light (700–780 nm) play a crucial role in phototropism, photosynthesis and light morphology during plant growth, respectively.

In contrast, ultraviolet (UV) and yellow-green light contribute little to plant photosynthesis [1]. Therefore, the means to increase the light content of blue light, red light and far-red light in the surroundings of plants to enhance plant photosynthesis, and hence promote plant growth has become a research hotspot in this field. Plant-growth LEDs and light-conversion films have been considered as two effective ways to increase the amount of beneficial light (blue, red and far-red) in the plant’s surroundings due to the modulation capabilities of their spectral components. A plant-growth LED is a phosphor-covered light-emitting diode. Due to the advantages of low energy consumption, high luminous efficiency, long life and adjustable wavelength, it has become the primary artificial light source in the process of plant growth and is widely used in indoor plant cultivation (IPC) so as to remove the limitation of insufficient light on plant cultivation [2].

In comparison, light-conversion films are mainly prepared by adding light-conversion agents to agricultural plastic films. It has the function of converting ultraviolet light and yellow-green light into blue light, red light and far-red light, which are strongly absorbed by plants so as to improve the light quality in the greenhouse and improve the comprehensive utilization rate of solar light energy. They are clean and environmentally friendly forms of a “light energy fertilizer” [3].

In both LEDs and light-conversion films, phosphors determine the light-conversion capacity, spectral range and stability of final products, which are key to effectively promote plant growth. Phosphors are mainly composed of the matrix and activator, whereby activator ions can enter the crystal lattice of the matrix as a luminescent center. After photoexcitation, they transit from the ground state to the excited state. Since the electrons in the excited state will transition from the excited state to the ground state, there will be a radiative transition in this process, resulting in luminescence.

In recent years, a large number of phosphors with excellent performance have been reported by researchers, such as SrMgAl_10−y_Ga_y_O_17_: Mn^4+^ (red phosphor) [1], Ca_1.1_Sr_0.9_SiO_4_: Ce^3+^ (blue phosphor) [4] and Ca_3_Al_2_Ge_3_O_12_: Eu^3+^ (far-red phosphor) [5]. However, phosphors with high luminous efficiency, good thermal stability and a high degree of match between the emission spectrum and plant pigment absorption spectrum are relatively rare. This is especially true for phosphors that can meet the requirements of different plants or the different growth periods of the same plant, as different kinds of light absorption are almost absent. Therefore, in this review, on the basis of reviewing the latest progress of phosphors used for plant growth, we discuss in detail the current regulatory strategies for the luminescent properties of phosphors used for plant growth. We believe that this review work will be helpful to explore more luminescent materials with excellent properties for plant growth (Figure 1).

## 2. Influence of Light on Plant Growth

Light, as an environmental signal and energy for photosynthesis, plays a crucial role in the process of plant growth. Only under the condition of light can plants carry out photosynthesis to synthesize and produce the required organic substances. In a review paper published in 2018, Pattison and colleagues suggested that plants might use several photoreceptors to direct their growth through light and control photosynthesis. These include phytochrome, carotenoid, chlorophyll, phototropic pigment and cryptochromes. Light is used as a fuel for photosynthesis and to control a plant’s form, along with delivering signals to guide fruit shape, leaf expansion and even color [6].

The main absorption range of plant photosynthesis is 300–800 nm, but not all light wavelengths can be used by plants, which are mainly determined by the effective characteristic absorption of plant photosynthetic pigments. The photosensitive pigment is mainly affected by red (660 nm) and far-red (730 nm) light. It dominates plant growth, development and differentiation processes, and includes P_R_ and P_FR_. These two photosensitive pigments with different absorption spectra can switch states between each other. The P_R_ is the biologically inactive state, which converts to the P_FR_ subtype by a strong absorption peak of 660 nm for red light, while the P_FR_ is the biologically active state, which reverts to the P_R_ state by capturing far-red light at a center of approximately 730 nm [7].

In contrast, cryptochromes and photochromes mainly require light from the blue region (400–490 nm), which affects the accumulation of plant biomass, is conducive to the growth and development of roots, and also has an important influence on the growth of plant stems [8]. Chlorophylls and carotenoids respond sensitively to blue (400–500 nm) and red (600–700 nm) light and are essential for plant growth and development [9]. Therefore, in the solar spectrum, blue, red and far-red light are particularly important for the growth and development of plants, which can effectively accelerate the growth rate of plants and improve the yield and quality of plants. Moreover, a small amount of UV light can improve plant resistance. Yellow-green light (490–550 nm) is also a form of photosynthetically active radiation, but it is generally believed that this light band contributes little to plant growth due to its low absorption by chloroplasts.

## 3. Fundamental Theory of Phosphors Luminescence

Inorganic phosphor materials are a type of photoluminescent material that can convert absorbed excitation light energy into emitted light for radiation. The basic composition of a typical inorganic phosphor material involves a matrix and an activator. In some cases, traces of other impurity ions (e.g., sensitizers, charge compensators) are introduced to enhance the luminescent properties of the materials [10]. The matrix, which is the main body of the phosphor, provides a suitable substitution site and lattice environment for the activating ion and consists mainly of inorganic compounds such as oxides, nitrides and sulfides. The activator is an optically active impurity doped in the matrix, usually in a small amount. The electron transition of the activator ion is crucial for producing luminescence. Activator ions that can be used as a luminescence center generally include rare earth elements, transition metal elements, etc. [11,12,13].

The fundamentals regarding the luminescence process of phosphor are commonly divided into two types: (1) the activator absorbing energy from the ground state to the excited state, and then through the form of light radiation back to the ground state, during which there is also a part of the form of heat generation through the non-radiative transition back to the ground state [14] and (2) the participation of a sensitizer, which absorbs energy from the ground state to the excited state and then transfers the absorbed energy to the activator through a form of energy transfer. The activator transfers the energy from the high energy level state to the low energy level state and then releases the energy back to the ground state in the same form as the previous class. This type of luminescence is often found in phosphors where the excitation energy absorption of the activator is relatively weak, and the energy transfer between the sensitizer and the activator is used to enhance the luminescence properties [15].

Quantum theory states that the electrons in an atom are in a state of rotation around the nucleus. The electrons can only rotate in a specific orbit, where no energy is produced. Atoms usually have many orbitals, and electrons in different orbitals have different energy states and form different energy levels. With a certain amount of energy, the electrons transition between different energy levels to achieve a change in luminescence [16].

## 4. Brief Description of Techniques for Improving Plant Growth

As the source of all lives, light plays an irreplaceable role in plant growth, such as germination, growth, flowering and fruiting [17]. Different wavelengths of light can lead to different effects on plant growth. In general, blue light favors photosynthesis, red light affects the phototropism of plants, while far-red light controls the entire life cycle of plants from germination to maturity. Therefore, increasing the light content of blue, red and far-red light in the plant’s surroundings will enhance plant photosynthesis, thereby promoting plant growth [18]. The selective absorption of light from plants makes it possible to control plant growth artificially. With the increasing demand for crop yield, the traditional methods of enhancing crop yields through chemical fertilizers and pesticides are no longer compatible due to their polluting nature. The technology of promoting plant growth and development by artificially increasing the content of beneficial light has been developed rapidly and has become one of the hot research fields in facility agriculture [19]. This can be roughly divided into plant-growth LEDs and light-conversion plastic films.

### 4.1. Indoor Plant-Growth LED

The traditional agricultural production methods are often affected by extreme weather conditions such as frost, hail, drought, fog and floods, making it difficult ensure crop yields and quality, and severely constraining economic and social development. With the improvement of the human living standard, the demand for a healthy life and a green environment is urgent day by day. However, the traditional method of increasing crop yields through fertilizers and pesticides will bring serious long-term pollution, which is no longer suitable for the needs of plant growth development. Therefore, with the development of modern agriculture, indoor plant cultivation has attracted more and more attention [20,21].

As the “engine” of plant growth, light plays a vital role in indoor crop cultivation [22]. The light sources used in traditional plant cultivation are mainly fluorescent, high-pressure sodium, incandescent and metal halide lamps. However, these light sources not only mismatch the photosynthetic spectra of plants, but also have high costs and a short lifespan. In comparison, phosphor-converting LEDs (pc-LEDs) are a new light source with high energy efficiency and environmental protection advantages. More importantly, the spectral composition of the pc-LEDs can be adjusted to better match the photosynthetic spectra of plants, which makes LEDs develop rapidly in the field of plant lighting [23]. LED lighting has been found to affect morphological or physiological changes in plants [24].

Figure 2 shows the configuration of the pc-LED. A pc-LED is a phosphor-covered light-emitting diode that emits light by combining blue or near ultraviolet (NUV) chips and phosphor [25]. Therefore, the final spectral range can be adjusted through the design of fluorescent powders to provide a better suit for plant photosynthesis. Table 1 shows recent research on phosphors for LEDs for indoor plant cultivation. The table shows that most of the reported plant LEDs are red phosphors and far-red phosphors, with relatively little research on blue phosphors. Nowadays, the high requirements for phosphor luminous intensity, thermal stability and spectral match have led to the rapid development of research into phosphors for LEDs on plant growth.

### 4.2. Light-Conversion Films

Light-conversion film is a functional film fabricated by adding light-conversion materials to a plastic film. It converts ultraviolet and yellow-green light from sunlight into blue, red and far-red light, which are strongly absorbed by plants. It can improve the quality of light in greenhouses and enhance the comprehensive utilization of solar energy, being a clean and environmentally friendly “light energy fertilizer” [37].

As a result of the loss of available arable land, increasing environmental pollution and the decline in food crops, functional films with specific functions are taking over the film industry. The light-conversion film adds the function of light-conversion in comparison to traditional agricultural film. It can effectively improve the photosynthetic efficiency of plants, increase their resistance to pests and reduce the use of pesticides. Compared with the plant-growth LED, it does not require any high-cost equipment, power and other inputs, in line well with the actual situation of Chinese agricultural development, and has excellent potential for development. Light-conversion agents are the core materials for light-conversion films to realize their light-conversion function. The properties of light-conversion films vary with the type of light-conversion agents. According to the material properties, it can be roughly divided into: rare earth inorganic compounds; rare earth organic complexes; inorganic luminous materials and organic fluorescent dyes [38].

Light-conversion film, a new-fangled type of agricultural material, has garnered significant interests in recent times owing to its potentials for enhancing light utilization efficacy, ensuring adequate light conditions for plants, and augmenting crop yield and quality. Consequently, the film has become the focal point of considerable research and investigations. However, there are still many deficiencies in the efficiency, cost, compatibility with agricultural film, light match with plant pigment absorption and light transmittance, which significantly limit the application of multifunctional agricultural films. Therefore, it is particularly important to develop new light-conversion agents with more excellent properties to promote the development of functional agricultural films.

## 5. Development of Phosphors for Plant Growth

### 5.1. Red Phosphor

Chlorophyll and photosensitive pigments are the primary photosynthetic pigments of plants, which have strong absorption of red light. Red light had significant effects on fresh weight, dry weight, stem length, leaf and leaf area [39]. Karimi et al. investigated the effects of red light on the physiological, morphological and phytochemical properties of goldenrod and found that with the increase in red light content, the leaf area and root elongation of plants increased, and the total fresh weight and dry weight of plants increased gradually. This is because red light induces cell division and cell expansion [40]. The principal activator ions commonly used in red phosphors for plant growth are Eu^3+^, Eu^2+^, Mn^4+^, etc. The luminescent characteristics and luminescent principles of these activator ions are described below.

#### 5.1.1. Eu^3+^-Doped Phosphor

The luminescence of Eu^3+^ belongs to the 4f–4f transition. Because its 4f orbital is shielded by 5s^2^5p^6^ orbital in space, the matrix crystal field has little influence on the spectrum of Eu^3+^-activated phosphor, so the luminescence is relatively stable. Since both the ground state energy level ^7^F_0_ and the excited state energy level ^5^D_0_ of Eu^3+^ are nondegenerate (J = 0), the emission and excitation transitions between the ^5^D_0_ and ^7^F_0_ energy levels can be used to determine the symmetry of the lattice sites occupied by Eu^3+^ ions [41]. Figure 3 shows a schematic diagram of the energy levels of Eu^3+^. Under near-ultraviolet or blue light irradiation, the excited electron transits from the ground state to excited state. Then, it returns to the ^5^D_0_ energy level after a non-radiative transition and achieves luminescence by a radiative transition from the ^5^D_0_ energy level to the ground state ^7^F_J_ energy level (J = 0–4) [42,43]. The ^5^D_0_→^7^F_J_ (J = 2,3,4) transition produces emissions in the 614 nm, 654 nm and 704 nm bands, in the range of red light (600–700 nm) strongly absorbed by plants. In addition, the synthesis process of Eu^3+^-activated phosphors is relatively simple as they are prepared by calcination directly under air without a reducing atmosphere. Therefore, Eu^3+^-activated red phosphors are widely used in plant growth.

Wang et al. prepared Ca_9_MY_0.667_(PO_4_)_7_(M = Li, Na):Eu^3+^ phosphors by a conventional solid-state reaction process. The matrix belonged to the R3c-type space group structure, a typical β-Ca_3_(PO_4_)_2_-derived structure, where the Ca^2+^ ions can be distributed in five crystal degree points with rich and diverse symmetries [42]. As shown in Figure 4a, the sample produces characteristic red light emissions under near-ultraviolet and blue light excitation. The 614 nm emission from ^5^D_0_→^7^F_2_ and the 700 nm emission from the ^5^D_0_→^7^F_4_ transition have the highest intensity, mainly attributed to the fact that Eu^3+^ occupies the lattice position at the non-inverted center of low symmetry. As shown in Figure 4b, the fluorescence emission spectra of Ca_9_NaY_0.667_(PO_4_)_7_:0.12Eu^3+^ matched with the absorption wavelengths of plant photosensitive pigments (P_R_ and P_FR_), which could effectively promote plant growth [43]. Sivakumar’s team reported a Li_3_BaSrLa_3_(MoO_4_)_8_: Eu^3+^ phosphor. Figure 4c indicates the excitation spectrum of the phosphor, which has strong absorption in the 200–430 nm region due to the overlap of the MoO_4_^2−^ group and the O^2−^→Eu^3+^ charge transfer band. Additionally, the large distance between the sites occupied by Eu^3+^ makes it possible to achieve high-concentration doping of the phosphor. Therefore, the phosphor has a high quantum yield (92.6%). Figure 4d shows that the spectral profile of the prepared red LED closely matches the absorption of the photosensitive pigment (P_R_), which indicates that the red phosphor can be applied in plant-growth LEDs [44].

In addition to the above inorganic luminescent materials, Eu^3+^-doped rare earth organic complex materials also have good luminescent properties. Shoji et al. prepared transparent films with UV to red wavelength conversion by coating [Eu(hfa)_3_(TPPO)_2_] luminophores mixed with TDMPPO on commercially available plastic films (as shown in Figure 5a,b). The film significantly promotes the growth of plants. Compared to plants developed by uncoated films, plants developed by coated films showed an increase in height by 1.2 times and overall biomass by 1.4 times [45,46].

#### 5.1.2. Eu^2+^-Doped Phosphor

Eu^2+^ has a 4f^7^ electronic configuration with a spectral term of ^8^S_7/2_ in the ground state. The lowest excited state may consist of the inner layer of the 4f^7^ group or the 4f^6^5d^1^ group. Nevertheless, the energy of the 4f^6^5d^1^ group of Eu^2+^ ions are generally lower than those of the 4f^7^ group at room temperature. Thus, most of the Eu^2+^ ion-activated materials are characterized by f–d transitions, which exhibit broad spectra and high-intensity emissions [47,48]. In addition, the matrix lattice field can significantly affect the Eu^2+^ activation of phosphors because of crystal field effects and nephelauxetic effects, resulting in an energy variation of the 5d energy level. The energy of the 5d can be reduced by tuning the matrix lattice field so that the Eu^2+^ transition occurs over a wide wavelength range, achieving a color shift from the ultraviolet to the red light-emitting region [49,50].

Eu^2+^-activated phosphors have been extensively studied in plant growth. For instance, Xia’s team changed the crystal field strength by adjusting the doping amount of Rb, thus modifying the emission peak of (Rb_x_K_1−x_)_3_LuSi_2_O_7_: Eu^2+^ to make it emit more suitable for the red light range required for plant growth (as shown in Figure 6a) [51]. Lei’s team has prepared red phosphors (Li_2_Ca_2_Mg_2_Si_2_N_6_: Eu^2+^) for the first time under atmospheric pressure using a simple solid-state method. It shows a red emission peak at 638 nm and a full width at a half peak (FWHM) of 62 nm under blue light irradiation. The phosphor emission spectrum remains almost constant at different excitation wavelengths except for the luminous intensity, indicating the presence of only one Eu^2+^ crystallization site in Li_2_Ca_2_Mg_2_Si_2_N_6_. The authors coated Li_2_Ca_2_Mg_2_Si_2_N_6_: Eu^2+^ red phosphor onto blue LED chips (λ_em_ = 455 nm) to fabricate plant-growth LEDs, and the device’s emission spectrum matches well with chlorophyll’s absorption range (Figure 6b). The cabbages treated by the plant-growth LEDs had better quality by comparing the growth of cabbages after 10 days under the same conditions with white daylight LED irradiation (Figure 6c), demonstrating that plant-growth LEDs based on this red phosphor can promote plant growth [52]. Furthermore, Lei’s team mixed Sr_2_Si_5_N_8_:2%Eu^2+^ phosphor with polyethylene to produce light-conversion films by extrusion, pelletizing and blow molding. SEM results images show that the film structure is dense after the addition of the light-conversion agents, leading to an increase in the mechanical properties of the film. The fluorescence spectrum shows that the light-conversion films convert blue-violet light to red light. Through simulation experiments, the biomass and quality of cabbage covered by the light-conversion films were found to be enhanced by comparison with ordinary agricultural films (Figure 6d), effectively promoting plant growth [53].

#### 5.1.3. Mn^4+^-Doped Phosphor

Mn^4+^, a transition metal ion, is frequently doped in various matrices as a red emission activator ion. The Mn^4+^ ion belongs to the d^3^ electron configuration, similar to the 5d electrons of Eu^2+^. Electrons in the d^3^ orbitals are also exposed, so the optical properties of Mn^4+^-doped materials are also strongly influenced by the crystal field environment [54]. The Mn^4+^ ion is normally stable in the octahedral position of the solid and emits red light between 600 and 700 nm when excited by two strong and broad excitation bands (^4^A_2g_→^4^T_1g_(^4^F) and ^4^A_2g_→^4^T_2g_). The emission is produced by the energy level transition of ^2^E_g_→^4^A_2g_. Since the lowest energy state (^2^E(t_2_^3^)) hardly changes in different crystal fields, the luminescence properties are very similar between different bodies but strongly depend on the covalency of the Mn^4+^ ligand bond [55,56]. The red light produced by the Mn^4+^-doped phosphor is an ideal match for the red light required for plant growth, which has potential applications in the field of plant-growth LEDs.

As shown in Figure 7a, Deng et al. reported a red light-emitting phosphor (BaZrGe_3_O_9_: Mn^4+^ (BZGO)). The main body of BZGO is a hexagonal structure which belongs to the P6(_)c2 space group. Its crystal structure is composed of two octahedrons of BaO_6_ and ZrO_6_. Mn^4+^ will replace Zr^4+^ due to the similar ionic radius and valence state. Under UV excitation, the red emission is in the wavelength range of 600–800 nm, which matches the absorption range with chlorophylls A, chlorophylls B and photosensitive pigments (P_R_) (Figure 7b). The emission intensity of BaZrGe_3_O_9_:Mn^4+^ at 150 °C is 81.7% of that at room temperature, showing excellent thermal stability. A red plant-growth LED was prepared using the phosphor, and the growth rate of garlic and corn was significantly improved under its irradiation. Since 3d→3d belongs to a cosmically forbidden transition, most Mn^4+^-doped phosphors have a relatively low excitation emission intensity [22]. In order to solve this problem, Wang et al. found that the red emission of Sr_4_Al_14_O_25_:0.01Mn^4+^ increased by 60% through the incorporation of Ga^3+^. This is mainly because the Ga–O bond is strongly covalent and can significantly alter the overlap of the 2p orbitals in the M–O–Mn bond, distorting the MnO_6_ octahedron and disrupting the cosmically forbidden d–d transition in Mn^4+^. They mixed the phosphors with polydimethylsiloxane to prepare the light-conversion film, and the subsequent chlorella growth experiment proved that the light-conversion film has great potential in promoting plant growth (Figure 7c) [21]. Wang et al. synthesized a red-luminescent ceramic (Mg_2_TiO_4_:Mn^4+^) using a high temperature solid-state reaction method. Compared with phosphor powder, luminescent ceramics have a higher thermal conductivity, which can effectively dissipate heat and reduce the influence of temperature on luminescence intensity. Under the excitation of 465 nm, luminous ceramics of Mg_2_TiO_4_:Mn^4+^ emits red light with a peak wavelength at 658 nm, which matches well with the wavelength required for photosynthesis [57].

### 5.2. Far-Red Phosphor

Light quality has a regulatory effect on plant photosynthesis. Plants perceive changes in light quality through different types of photoreceptors, among which photosensitivities mainly absorb red and far-red light. When R:FR is low, plants will produce a series of shade avoidance reactions, thus affecting plant growth [58,59,60]. Kalaitzoglo et al. found that increasing R:FR in plant-growth LEDs can negatively affect the growth and early fruit yield of tomato seedlings. The main reason for this is that the lack of far-red light reduces the total leaf area, which in turn leads to a reduction in total plant light absorption and plant dry mass [61]. Far-red light has an important influence on plant growth. In recent years, research on far-red phosphors for plant growth mainly doped with oxides of Mn^4+^ and Cr^3+^ have developed rapidly. Under UV excitation, Mn^4+^-doped red phosphors emit primarily red light around 660 nm. Since its luminescence properties are closely related to the crystal field environment of the matrix, the researchers have prepared a series of Mn^4+^-doped phosphors with far-red light emission by modulating the crystal environment. For example, Ye’s group has successfully synthesized a series of Mn^4+^-activated SrLa_2_Al_2_O_7_ phosphors using a high-temperature solid-phase method. Excited by 365 nm, the phosphor of SrLa_2_Al_2_O_7_: Mn^4+^ emits a far-red light with a peak at 731 nm, which matches well with the absorption band of the photosensitive pigment P_FR_ (Figure 8a) [29]. In addition to finding suitable substrates for Mn^4+^-doped far-red emission phosphors, cationic substitution can also be used to adjust the luminescence spectrum. Zhou et al. substituted Ga^3+^ for the Al^3+^ in the red phosphor of SrMgAl_10-y_Ga_y_O_17_ doped with Mn^4+^, as shown in Figure 8b. The luminescence spectrum produced an obvious red shift, which was mainly related to the energy reduction of the ^2^E level caused by the weak electron cloud rearrangement effect brought by the substitution [1,62].

Recently, the transition group metal Cr^3+^, along with Mn^4+^, has also been the focus research on far-red phosphor-activated ions. The electronic configuration of Cr^3+^ is 3d^3^, with wide absorption in its transition energy levels of ^4^T_2_→^4^A_2_ and ^4^T_1_→^4^A_2_, which can be well-matched to the UV chip. The characteristic emission of Cr^3+^ exhibits spin-forbidden narrow-band red emission (^2^E→^4^A_2_) and spin-allowed broadband far-red emission (^4^T_2_→^4^A_2_), making Cr^3+^-doped phosphors a potential application for plant growth [63]. The strength of the crystal field determines the dominance of the narrow-band red light emission and broadband far-red light emission of Cr^3+^ phosphors, so the choice of matrix is particularly important for Cr^3+^-doped phosphors [64]. Huy’s team has successfully synthesized Cr^3+^-doped BaMgAl_10_O_17_ (BAM) narrow-band red phosphors with a hexagonal β-aluminum oxide structure using the sol–gel method. Its strong narrow-band emission (FWHM ~ 4 nm) peaked at 695 nm. Cr^3+^ doping replaces Al^3+^ in the matrix at a distorted octahedral position with a strong crystal field (D_q_/B > 2.3), resulting in a narrow-band red emission from the phosphor of ^2^E→^4^A_2_ (Figure 8c) [65]. In contrast, Xia’s team produced a broadband near-infrared emission at 650–1350 nm by doping Cr^3+^ into the weak octahedral coordination crystal field of the lead-free metal halide bis-chalcogenide of Cs_2_Ag_1−x_Na_x_InCl_6_ [66]. Based on the ability of the crystal field strength to modulate the luminescence of Cr^3+^-doped phosphors, Li’s team has achieved a modulation of the crystal field strength at the center of the luminescence by a co-substitution of Gd^3+^ in Gd_3_Ga_5_O_12_ with Y^3+^ and In^3+^. As the co-substitution proceeds, D_q_/B gradually decreases, resulting in a redshift of the spectrum and a broadening of the FWHM. The electroluminescence (EL) spectrum of a pc-LED made using this phosphor is shown in Figure 8d. The EL spectrum covers well with the light required for P_R_, P_FR_, chlorophyll B and phycocyanin during plant growth. Light tests on foliage green plants revealed that the best quality foliage green plants were grown in a R+NIR light environment, thus resulting in the positive application of Gd_3_Y_0.5_In_0.5_Ga_5_O_12_: Cr^3+^ phosphors to plant growth [64].

**Figure 8 nanomaterials-13-01715-f008:**
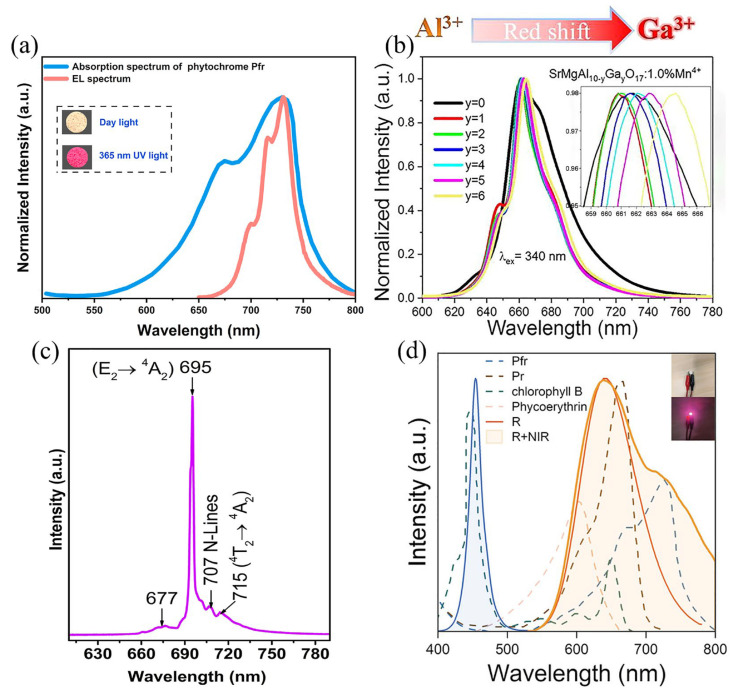
(**a**) Comparison of the absorption spectrum of phytochrome P_FR_ and EL spectrum of pc-LED by using SrLa_2_Al_1.994_O_7_:0.006Mn^4+^ phosphor. (Reprinted from ref. [29]. Copyright of Elsevier, 2023). (**b**) The normalized PL spectra under 340 nm excitation and the insets are an enlarged part of the peaks in the wavelength ranging from 658 to 668 nm. (Reprinted from ref. [1]. Copyright of Elsevier, 2020). (**c**) PL spectra measured under excitation wavelength of 405 nm of the BAM:1%Cr^3+^ phosphor. (Reprinted from ref. [65]. Copyright of Elsevier, 2020). (**d**) Infrared part of the LED package spectrum with the absorption curve of chlorophyll B, P_R_, P_FR_ and phycoerythrin. (Reprinted from ref. [64]. Copyright of John Wiley and Sons, 2022).

### 5.3. Blue Phosphor

The whole life cycle of plants is strongly influenced by the light environment, especially the light involved in photosynthesis and photophore reaction. Blue light is widely involved in plant growth and development processes, such as photomorphogenesis and leaf expansion. In a study by Li et al., the chlorophyll A, chlorophyll B, carotenoids and total photosynthetic pigments in the leaves of plants irradiated with blue light were 1.3 times higher than in control plants. The growth of the leaves, as well as the rootstock of the plant, is remarkable [23,67,68]. Currently, there are relatively few studies on blue phosphors for plant growth, with only a few Eu^2+^- and Ce^3+^-doped phosphors. Regarding Eu^2+^-doped luminescent materials, their use in red phosphors for plant growth has been described above as having a wide range of applications. Since the 5d energy level of Eu^2+^ is susceptible to the external crystal field environment, the wavelength position of its emission band is very matrix-dependent, allowing for luminescence modulation from the near-UV to the far-IR [69]. Regarding Eu^2+^-doped blue phosphors for plant growth, Guo’s team used a high-temperature solid-phase method to synthesize an Eu^2+^-doped luminescent material (Na_2_BaSr(PO_4_)_2_). The phosphor emits a blue light at 400–500 nm, with a main peak at 428 nm when excited by UV light. The blue region has a spectral similarity index (SR) of 92% with chlorophyll A, which is highly compatible with the absorption spectrum of plant chlorophyll A in the blue light region [36].

Among the rare earth luminescent ions, Ce^3+^ has the simplest 4f^1^ electronic configuration. The luminescent characteristics of Ce^3+^ are caused by an optical transition between the 4f^1^ ground state and the 5d^1^ excited state. The 4f^1^ ground state configuration splits into two energy levels (^2^F_5/2_ and ^2^F_7/2_) with the number of 5d energy levels depending on the symmetry of the crystal field. The most significant effects on the interaction of Ce^3+^ with the matrix lattice are the crystal field splitting and nephelauxetic effect, so the position of the excitation and emission bands is closely related to the matrix lattice (i.e., crystal structure, crystal field environment and nephelauxetic effect). Ce^3+^ emits near-UV to yellow light, and is mostly used as a blue light source in plant growth [70,71]. For example, Lian’s team developed the Ce^3+^-doped alkaline earth silicate phosphor (Ca_m_Sr_2-m-n_Ba_n_SiO_4_: Ce^3+^, Li^+^ (CSBS: Ce^3+^)). The spectral modulation was achieved by varying the composition of Sr and Ba as a proportion of the matrix. A redshift of the spectrum from 417 nm to 438 nm was produced from Ca_2_SiO_4_ to Sr_0.3_Ca_1.6_SiO_4_, and a blueshift of 401 nm was produced when the matrix was Ba_2_SiO_4_. Different plants have slightly different absorption spectra. This spectrally tunable phosphor has great potential for use in plant-growth LEDs and light-conversion films [72].

### 5.4. Red and Blue Composite Phosphors

Although specific wavelengths of red and blue light are absorbed by plant photoreceptors and have a positive effect on plant growth, individual blue or red light irradiation not only cannot meet their needs in the actual process of plant growth, but also has some negative effects [73]. Research has found that plants exposed to monochromatic red light can develop “red light syndrome.” This is mainly due to dysfunctional photosynthesis, such as low photosynthetic capacity and slow stomatal response, resulting in reduced plant biomass, leaf area, leaf number, chlorophyll content and stem elongation [74,75,76]. This “red light syndrome” can be reversed by adding blue light. Miao et al. studied the effects of monochromatic red and red-blue composite light on chloroplast ultrastructure, photosynthesis and nutrient accumulation in cucumber seedlings. The results show that by adding blue light, an increase in chloroplasts, larger starch grains and thicker leaves can be obtained, which will directly benefit leaf photosynthetic performance and thus maintain dry matter accumulation [77]. Therefore, the study of red and blue composite phosphor is of great significance for plant growth.

#### 5.4.1. Single-Ion-Doped Red and Blue Composite Phosphors

Eu^2+^ is the best-known and most widely used of a single-ion-activated multi-color phosphor. The 5d→4f energy transition of Eu^2+^ is closely related to the crystal environment of the occupied sites, so single-phase-doped multi-color phosphors can be obtained by combining multiple cation sites in a single matrix with different crystal field environments [78,79]. Guo’s group has studied the Eu^2+^-activated phosphors Ba_3_GdNa(PO_4_)_3_F (BGNPF). As shown in Figure 9a, the matrix Ba_3_GdNa(PO_4_)_3_F has three different Ba^2+^ sites, including Ba1 with six O coordination, Ba2 with two F coordination to five oxygen atoms, and Ba3 with two F atoms coordination to six oxygen atoms, which are partially substituted by Eu^2+^. Emission spectra and liquid helium temperature lifetimes demonstrate that Eu^2+^ enters the three Ba^2+^ crystal sites and produces emissions at 436 nm, 480 nm and 640 nm, originating from the 4f^6^5d^1^→4f^7^ transition. Electroluminescence spectra of an LED fabricated by combining this phosphor with a near-UV chip include two broadband emission peaks in blue (472 nm) and red (608 nm), respectively. It has a high degree of match with the absorption spectra of chlorophyll and carotenoid (Figure 9b) [3]. In addition to the blue-red dual emission produced by Eu^2+^ at different sites, the red light emission of Eu^3+^ is also present in some Eu^2+^-doped phosphors. A single-ion-realized red-blue dual-emission phosphor can also be realized by combining it with the blue-emitting Eu^2+^ in this system. For example, Xia et al. synthesized a blue-red dual-emission dual-phase phosphor (Na_3_La_2_(PO_4_)_3_:xEu (Na_3_La(PO_4_)_2_/LaPO_4_:xEu)) by a high-temperature solid-state reaction method. The main body of the phosphor is a composite material composed of Na_3_La(PO_4_)_2_ and LaPO_4_ crystals. As seen in Figure 9c, the phosphor shows blue light emission at 422 nm and red light emission at 621 nm under UV excitation. This is mainly due to the presence of both Eu^2+^ and Eu^3+^ in the phosphor [80].

#### 5.4.2. Double-Ion Co-Doped Red and Blue Composite Phosphors

In addition to the doping of a single luminescent center, the simultaneous doping of the matrix with different activated ions enables the production of red and blue dual-emission phosphors. For example, Zhang et al. doped two luminescent centers (Ce^3+^ and Mn^2+^) into the matrix Na_4_CaSi_3_O_9_ (NSCO). The phosphor exhibits a blue-red double emission with the excitation of 336 nm. The blue emission originates from the 5d→4f transition of the Ce^3+^, and the red emission originates from the ^4^T_1_(^4^G)-^6^A_1_(^6^S) forbidden spin transition of Mn^2+^. The emission spectrum of the phosphor has a superior overlap with the absorption spectrum of chlorophyll, as shown in Figure 9d. It can be seen that the light emitted by the phosphors can be well-utilized by plants and has potential applications in the field of plant growth [81]. Moreover, Gong et al. investigated the co-doping of Eu^2+^ and Mn^2+^ to achieve red and blue dual emissions. (Na_2_Mg_2_Si_6_O_15_ was chosen as the matrix). Eu^2+^ randomly occupies the 6-ligand and 7-ligand of Na^+^ lattice sites in the matrix, while Mn^2+^ occupies one of the octahedral Mg2 sites to produce the red light emission. On the other hand, a significant spectral overlap can be observed between the Eu^2+^ emission band and the Mn^2+^ excitation transition. This demonstrates that Eu^2+^ can also transfer a part of its energy to Mn^2+^, effectively enhancing its forbidden energy level (^4^T_1_-^6^A_1_) transition. Figure 9e shows the emission spectrum of the phosphor under 365 nm UV excitation, with the blue and red emission peaks matching the photosynthetic action spectrum (PAS) of the plant chlorophyll [82].

## 6. Luminescence Regulation Strategy of Phosphors

To better cater to the light requirements of plant growth, phosphors for plant growth fulfill the following criteria. (1) The excitation spectrum of the phosphor utilized in light-conversion films should exhibit a significant degree overlap and strong absorption with the ultraviolet or yellow-green light in the solar spectrum. The excitation spectrum of the phosphor applied in plant-growth LED should demonstrate a strong correlation with n-UV light or blue light as well as exhibit high absorption intensity. (2) The emission spectrum of the phosphor should have a high spectral match to the absorption spectrum of the plant. The phosphor must not only emit the blue, red and far-red light required for plant growth, but also possess a sufficiently high emission intensity. (3) Possessing a high quantum efficiency results in efficient luminescence with minimal energy loss. (4) The phosphor must exhibit chemical stability in humid air, bright light and other environmental conditions. Furthermore, the phosphor must exhibit exceptional thermal stability to ensure optimal performance in plant-growth LED applications at temperatures of up to 150 °C. According to the aforementioned, it is essential to adjust and improve the luminescence performance of phosphors. Benefiting from the tunability of the rare earth ion/transition metal ion, specific strategies can be employed to optimize the luminescence properties of the phosphor, such as spectral position, half-peak width and thermal stability. The relevant modulation strategies are described below.

### 6.1. Adjustment of the Spectral Position

The light absorption of plants is dependent on the type of photoreceptors they possess. These photoreceptors absorb various wavelengths of light, resulting in plants having distinct requirements for specific wavelengths of light. Moreover, plants have different light requirements at different stages of growth, making it crucial to develop phosphors with adjustable spectra for optimal plant development. As mentioned above, benefiting from the close link between the energy levels of rare earth ions/transition metal ions and the crystal environment, the crystal field strength can be modified by structural and compositional design to modulate the excitation and emission wavelengths of the activator. The specific strategies are summarized below.

#### 6.1.1. Strategy for Group Substitution

Substituting an element in the matrix composition by another element of a different ionic radius alters the lattice environment surrounding the activating ion, resulting in a shift of its emission peak. Matrix replacement can be divided into three types: cationic substitution, anionic substitution and chemical unit co-substitution. Among these, cation substitution can be further classified into matrix cation substitution and cation substitution within anionic coordination polyhedra. For example, Wang’s group found that the emission spectrum of Rb_x_K_2-x_CaPO_4_F: Eu^2+^ changed from the red emission (665 nm) of K_2_CaPO_4_F: Eu^2+^ to the cyan-blue emission (487 nm) of Rb_2_CaPO_4_F: Eu^2+^ by replacing K^+^ with Rb^+^ (as shown in Figure 10a). The blue shift in the emission spectrum can be attributed to the large lattice expansion of the Rb_x_K_2-x_CaPO_4_F: Eu^2+^ as the concentration of Rb^+^ with a large radius increases. The large lattice expansion results in an increase in the average bond length of the Eu–O bond, a decrease in the crystal field strength and a reduction in the cleavage of the 5d electron, ultimately leading to an increase in the emission energy [83]. However, Gu et al. discovered that a spectral shift was also observed in Mn^4+^-doped phosphors (SrMgAl_10−y_Ga_y_O_17_) when Al^3+^ was replaced by Ga^3+^ in the anionic group. The nephelauxetic effect is the primary factor contributing to the spectral shift. The introduction of Ga^3+^ resulted in a weakened electron cloud expansion effect between the bonds, leading to a reduced Racah parameter B value, which subsequently caused a decrease in the energy of ^2^E_g_→^4^A_2_ transition, thus resulting in a redshift phenomenon [1]. In summary, the crystalline field strength of the activating ion and the nephelauxetic effect are simultaneously altered by cation substitution, with the final change in spectral position guided by the stronger of these effects.

Anion substitution is another strategy for modulating the luminescent properties of phosphors. The covalence of the crystal is greatly influenced by electronegativity difference between the coordination anions, and the nephelauxetic effect is closely related to the covalence of the crystal. Therefore, the substitution of different electronegativity anions can alter the nephelauxetic effect, modify the energy gap between the lowest energy level of 5d orbitals, ultimately resulting in an adjustment of emission energies and spectral position. The order of electronegativity of several common anions is C^4−^ <N^3−^ <Cl^−^ <O^2−^ <F^−^. Gong et al. achieved controlled green-to-blue regulation in the Ba_3_Ca_2_(PO_4_)_3_F: Eu^2+^ system using an anion substitution strategy of Cl^−^ for F^−^ [84]. Similarly, the introduction of the N^3−^ anion in the Ce^3+^-doped Ca_3_Sc_2_Si_3_O_12_ phosphor resulted in an emission redshift due to the reduction in energy difference between the lowest 5d excited state and ground state of Ce^3+^ [85]. In addition, luminescence can be regulated by the co-substitution of chemical units. As shown in Figure 10b, Han Tao’s group substituted [Ca^2+^–Ge^4+^] with [Lu^3+^–Ga^3+^] in the phosphor of Ca_3−x_Lu_x_Ga_2+x_Ge_3−x_O_12_: Cr^3+^, resulting in a shift of the PL peak position from 766 to 803 nm. On the one hand, the co-substitution of [Lu^3+^–Ga^3+^] leads to a decrease in lattice parameters and cell volume, and the contraction of the lattice usually leads to a strong crystal field splitting. On the other hand, the co-substitution reduces the band gap, leading to a downward shift in the energy level of the lowest position of Cr^3+^ (^2^E→^4^T_2_) and the redshift of the luminescence spectrum [86].

#### 6.1.2. Strategy for Activating Ion Concentration and Phosphors Concentration Regulation

For matrix materials containing multiple cation sites, differences in the lattice environment among these sites cause the same luminescent center to exhibit different luminescence. Additionally, changing the doping concentration in certain systems can impact the lattice position occupied by the activator ion, resulting in the change of luminescent color. In the example of Sr_3_Sc_4_O_9_:xEu phosphor, the matrix Sr_3_Sc_4_O_9_ contains three types of Sr^2+^ sites, Sr1O_6_, Sr2O_9_ and Sr3O_1_, and an Sc site that can be replaced by Eu^2+^. The selective site occupation of Eu^2+^ ions in the matrix occurs as the Eu^2+^ doping concentration increases. The least energy is required to enter the 6-ligand cation site, so the Eu^2+^ ion preferentially occupies the 6-ligand Sr and Sc sites and exhibits only one red light emission band (Figure 10c). As the x value increases, the Eu^2+^ ions begin to occupy the Sr2 and Sr3 sites, showing full spectral emission. In higher doping concentrations, the Eu^2+^ cation only occupies the Sr1–3 sites, and the long-wave NIR emission peak from the Sc^3+^ site disappears. This is because Eu^3+^ is a better match to Sc^3+^ in terms of radius and charge, so at higher concentrations, the Eu ions that enter the Sc^3+^ site are in the trivalent state to maintain the stability of the overall crystal structure [87]. Furthermore, similar phenomena can be observed in phosphor systems such as Sr_3_Ga_2_Ge_4_O_14_: Cr^3+^ [88], LiYGeO_4_: Tb^3+^ [89] and Li_2_Mg_3_TiO_6_: Cr^3+^ [90]. Phosphors must ultimately be dispersed in specific types of polymers, such as poly(methyl methacrylate) (PMMA), poly(dimethylsiloxane) (PDMS), poly(vinylpyrrolidone) (PVP), poly(ethylene oxide) (PEO), etc., to make light-conversion films or plant-growth LEDs. The combination of phosphor and polymer can have a positive effect on the phosphor [91]. For example, by protecting the phosphor from environmental conditions or preventing aggregation, the fluorescent properties of the powder can be improved, which would otherwise lead to sudden extinction. It has been discovered that altering the phosphor concentration in the polymer affects the phosphor’s dispersion and interaction with the polymer, modifying the composite’s luminous color. As an illustration, Maria Luisa Saladino et al. discovered that the emission peaks of Ce^3+^: Y_3_Al_5_O_12_ (Ce:YAG)-PMMA composites shifted by 15 nm as the amount of Ce:YAG increased and that the clusters of Ce:YAG particles were more evenly distributed in composites with higher doping concentrations. The xrd diffraction pattern of the samples with different concentrations showed no change in the Ce:YAG peaks compared to the powder, proving that it was not the change in phosphor structure that caused the shift in the luminescence position. Further solid-state NMR tests showed that the addition of Ce:YAG particles strongly influenced the T_1𝜌_(C) values. The change in relaxation time demonstrates that electron donor interactions occur between the carboxy-oxygen lone pair and the surface yttrium or cerium ions, resulting in a shift in the emission spectrum and a more homogeneous composite [92]. The same phenomenon was observed in composites formed by the doping of the rare earth organic complex Eu(TTA)_3_(H_2_O)_2_ (TTA: thenoyltrifluoroacetone) in poly(ethylene oxide) (PEO), where the luminescence position of the composite shifted with the change in the molar ratio of the complex to the polymer [93].

#### 6.1.3. Crystal Phase Engineering

In some special cases, activator doping or component substitution can cause a change in the crystalline phase of the matrix. Different crystalline phases will naturally have different lattice environments, which results in luminescence modulation in the phosphor. For example, Prof. Lian’s team formed an additional K_2_BaCa(PO_4_)_2_ phase by introducing Ba^2+^ into K_2_Ca(PO_4_)F: Eu^2+^, and its content increased with Ba^2+^. The luminescence spectrum also gradually shifts from red emissions in the K_2_Ca(PO_4_)F matrix to cyan emissions in K_2_Ca(PO_4_)F, achieving tunable emission colors (Figure 10d) [94].

**Figure 10 nanomaterials-13-01715-f010:**
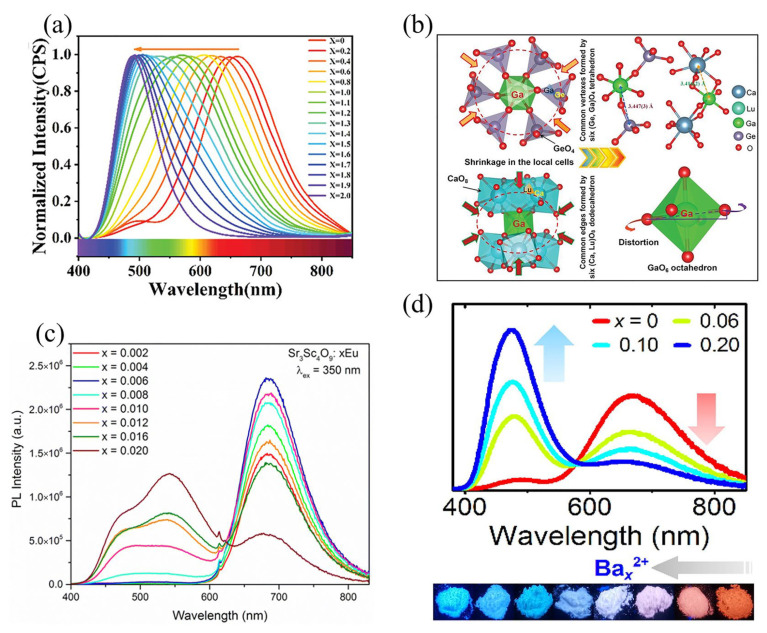
(**a**) Normalized emission spectra of Rb_x_K_2−x_CaPO_4_F: Eu^2+^ (0 ≤ x ≤ 2) excited at 380 nm. (Reprinted from ref. [83]. Copyright of John Wiley and Sons, 2021). (**b**) Schematic illustration of the first and second coordination environments. (Reprinted from ref. [86]. Copyright of John Wiley and Sons, 2021). (**c**) The PL spectra and of Sr_3_Sc_4_O_9_: xEu excited at 350 nm. (Reprinted from ref. [87]. Copyright of Royal Society of Chemistry, 2022). (**d**) PL spectra of K_2_BaCa(PO_4_): Eu^2+^ phosphors and the photographs of the samples under a 365 nm UV lamp. (Reprinted from ref. [94]. Copyright of Elsevier, 2023).

### 6.2. Adjustment of the Spectral Width

The broad blue and red light spectra have a better match for the absorption spectrum of the plant, providing a more effective support for plant growth and development. There are two main strategies for broadening the emission spectrum. One method is to dope the activating ion into multiple lattice sites in the matrix, allowing for different crystal field environments and resulting in a broad-spectrum emission from the combination of luminescence at different sites.

As shown in Figure 11a, the BaIn_2_(P_2_O_7_)_2_ (BIP) structure has two unequal [InO_6_] octahedral positions, both of which share angles with six [PO_4_] tetrahedra. The average bond lengths of In1-O and In2-O are 2.1311 Å and 2.2317 Å, respectively. BaIn_2_(P_2_O_7_)_2_:0.02Cr^3+^ exhibits a broad near-infrared emission in the range of 700–1400 nm when excitation occurs at 480 nm. The broadband NIR emission arises from the overlap of two spectral bands centered at 879 and 958 nm, originating from Cr^3+^ located in distinct [InO_6_] octahedra [95]. Similarly, broadband emission resulting from multi-location occupation was observed in Bi^3+^-doped phosphors. There are three types of Sr sites in the matrix Sr_3_Al_2_O_5_Cl_2_ that can be occupied by Bi^3+^ ions. It was determined that the Bi^3+^ ion exclusively occupied Sr1 and Sr3 positions. Due to the longer average bond length of the Sr1 site, broadband emission at 490 nm dominated by the Sr1 position is generated under excitation at 345 nm excitation, and 556 nm dominated by the Sr3 position is produced upon excitation at 376 nm [96].

Another strategy is to co-dope the activator with a distinct emission originating from other ions. The emission spectrum can be broadened by activating the energy transfer between the ions and other luminescent centers. In the process of energy transfer, other luminescent centers not only generate their own emissions, but also act as sensitizers to transfer energy to the activating ion. The prerequisite for achieving efficient energy transfer is that the emission spectrum of the sensitizer must have a certain degree of overlap with the absorption spectrum of the activator. For example, the emission spectrum of Eu^2+^ overlaps with the excitation spectrum of Mn^2+^ in the phosphor of KMg_4_(PO_4_)_3_: Eu^2+^, Mn^2+^, which offers the possibility of energy transfer from Eu^2+^ to Mn^2+^. Under excitation of 365 nm, the KMg_4_(PO_4_)_3_: Eu^2+^, Mn^2+^ phosphors exhibit a broad excitation band ranging from 250 to 425 nm and two broad emission bands peaking at 450 nm and 625 nm, which are attributed to the 4f-5d transition of Eu^2+^ and the ^4^T_1_→^6^A_1_ transition of the Mn^2+^ ion. As shown in Figure 11b, the FWHM value of the Mn^2+^ ion emission peak band in KMg_4_(PO_4_)_3_ gradually increases with increasing Mn^2+^ concentration. The energy transfer from Eu^2+^ to Mn^2+^ effectively broadens the emission spectrum within the red spectral region [97]. In addition, the Cr^3+^ ion has a broad absorption band throughout the UV–Vis region, and its excitation spectrum overlaps with most of the luminescent ions, such as Eu^2+^, Ce^3+^, Bi^3+^, Yb^3+^, etc. Lei’s group has studied Eu^2+^-, Cr^3+^-doped Mg_2_A_l4_Si_5_O_18_ phosphors. Mg_2_A_l4_Si_5_O_18_ provides sufficient sites for the incorporation of Eu^2+^ and Cr^3+^. The diffuse reflectance spectra (DRS) of Mg_2_Al_4_Si_5_O_18_: Cr^3+^ and the emission spectra of Mg_2_Al_4_Si_5_O_18_: Eu^2+^ exhibit spectral overlap in the range of 500 nm to 850 nm (Figure 11c), which indicates the possibility of energy transfer between the Eu^2+^ and Cr^3+^ ions [98]. The phosphor spectrum of Mg_2_Al_4_Si_5_O_18_: Eu^2+^ shows the characteristic 4f^6^5d→4f^7^ transition red emission of Eu^2+^ under 450 nm excitation. Excitation emission spectra with characteristic excitation/emission bands of Eu^2+^ and Cr^3+^ were clearly identified in Mg_2_Al_4_Si_5_O_18_: Eu^2+^, 0.01Cr^3+^, implying the successful construction of ultra-wide vis–NIR phosphors. Except for co-doping with other luminescent ions as sensitizers to increase the spectral emission range, doping activating ions in matrix with self-luminescent properties is also an effective means of achieving a broad-spectrum emission. For instance, Xie’s group synthesized Cr^3+^-doped SrHfO_3_ (SHO) by high-temperature solid-phase method. SHO emits near-infrared light in the range of 700–800 nm due to the presence of self-trapping excitons. Considering the similarity to an ionic radius, Cr^3+^ ions are preferred to replace Hf^4+^ ions in the octahedral position. As shown in Figure 11d, the emission band of SHO:0.005Cr consists of two parts when excited at 460 nm. The spectral emission at 700–850 nm originates from the autofluorescence of the SHO host, the emission at a peak half-height width of 190 nm at 1000 nm results from the transitions of ^4^T_2_→^4^A_2_ allowed by the Cr^3+^ spin [99].

### 6.3. Adjustments of Other Performance

Quantum efficiency is an essential indicator of a luminescent material’s light-conversion capability, reflecting the phosphor’s ability to convert absorbed energy into emitted light. The higher the quantum efficiency, the more efficient the light-conversion, resulting in less energy loss. There are several strategies to improve the quantum efficiency of phosphors, and all of the methods mentioned above have been reported to optimize quantum efficiency. For example, Wang et al. achieved a 16.67% increase in internal quantum efficiency (IQE) by a cation substitution strategy of Al to In in the Cs_2_KInF_6_: Cr^3+^ system [100]. Yan et al. reported Mn^4+^ and Bi^3+^ co-doped Gd_2_SrAl_2_O_7_ phosphors with a significant increase in quantum efficiency (79.9%) by energy transfer from Bi^3+^ to Mn^4+^ [101]. The main focus will be to introduce a different strategy: the flux strategy which aims to improve the quantum efficiency of the phosphors. Li et al. successfully prepared a series of yellow light-emitting samples (Ca_2_MgWO_6_: Bi^3+^) using a high-temperature solid-phase method. The addition of MgF_2_ as a flux in the preparation resulted in the enhancement of the luminescence properties by adjusting the lattice and morphology of the model. The X-ray diffraction pattern is shown in Figure 12a. With the increase in MgF_2_ concentration, the impurity phase of CaWO_4_ initially present gradually decreased. When the addition amount reached 2 wt%, the impurity phase disappeared, and the emission intensity of Bi^3+^ ions increased by 21.4%. The phosphor has a high quantum efficiency and absorption efficiency with maximum values of 42.1% and 89.1%, respectively [102]. Park et al. investigated the effect of different fluoride fluxes on the photoluminescence properties of Lu_2.94_Al_5_O_12_:0.06Ce^3+^, such as LiF, NaF and BaF_2_. The melting point of the flux is lower than the solid-state reaction temperature (1400 °C). This liquid phase will be present as a liquid phase during the reaction and acts as a facilitator of ion diffusion, thereby facilitating the formation of highly crystalline phosphors. The excitation-emission spectra show that the addition of the flux does not alter the shape and position of the phosphor excitation peak but only increases the intensity. However, the emission peak was blue-shifted, which was attributed to the addition of a flux. The addition of a flux increased the crystallinity of the phosphor and decreased the lattice defects around the Ce^3+^ ions, thus reducing the crystal field splitting [103].

Apart from quantum efficiency, excellent thermal stability is essential for phosphors used in plant-growth LEDs because the operating temperature can reach up to 150 °C. Nevertheless, the luminescence intensity of fluorescent materials generally decreases with the increase in temperature, which is called the thermal quenching phenomenon. Therefore, improving the thermal quenching performance of phosphors remains a major research focus. The thermal stability of luminescent materials is closely related to the matrix lattice. The high structural stiffness of the matrix lattice represents a weak lattice vibration, which essentially limits the non-radiative transition within the phosphor and enhances the thermal burst resistance of the phosphor. For example, You’s group has synthesized a series of Cr^3+^-activated LiABO_4_ (a = Al, Ga; B = Ge, Si) with a similar benzene ring structure. As shown in Figure 12b, the crystal structures of LiGaGeO_4_ and LiAlSiO_4_ are rigid and similar to that of a benzene ring. Due to the extensive structural stiffness, the emission intensity of LiGaGeO_4_:002Cr^3+^, LiAl_0.1_Ga_0.9_GeO_4_:0.002Cr^3+^ and LiAlSiO_4_:0.02Cr^3+^ at 423 K (150 °C) remains at 85%, 90% and 94% compare to that at 298 K (25 °C) (Figure 12c,d) [104]. In addition, some trap energy levels can be created in matrix through non-equivalent substitution. These trap energy levels can act as electron-capture centers to store the energy and release electrons at a high temperature. These electrons which return to the ground state can compensate for the loss of thermal bursts, thus improving the thermal stability of the phosphor. Bai et al. reported a new red-emitting phosphor K_2_MgGeO_4_: Eu^3+^ which exhibits zero thermal quenching. As shown in Figure 12e, With the increase in temperature, the emission intensity of the phosphor gradually increased. When the temperature rises to 300 °C, the intensity remains at 107.22% of the initial value at room temperature. The ICP-OES results revealed that the actual content of the potassium element was lower than the theoretical content, indicating the presence of potassium vacancies in the lattice of the doped sample. In addition, the increasing trend of oxygen concentration suggests the presence of interstitial oxygen atoms in the lattice. O_i_ is also demonstrated by the deconvolution peak at 533.76 eV in the high-resolution XPS spectrum. Due to the non-equivalent substitution of Eu^3+^ ions for K^+^, two kinds of defects are generated in the lattice. Under thermal activation, the electrons trapped by the defects are re-released to participate in the radiative transition, thus compensating for the energy loss caused by the non-radiative transition at high temperature, and finally achieving a zero thermal quenching performance of the phosphor [105].

## 7. Summary and Outlook

In this review, we systematically summarize the research progress of phosphors for plant growth. The main classifications are based on the central light-absorbing regions of plant photosynthesis, including red, blue, far-red and red-blue composite phosphors. The luminescent properties of the relevant phosphors are described from the perspective of the luminescent ions. In particular, several strategies to meet the requirements for phosphors are summarized in the direction of light-conversion films and plant-growth LEDs by adjusting the spectral wavelength, broadening the emission spectrum and improving quantum efficiency and thermal stability. Modifying the spectral position through component substitution in the matrix structure is a very effective method. Additionally, adjusting the concentration of the activator and altering the crystal phase to modify the crystal environment around the luminescent center is an alternative strategy for achieving spectral shifts. The selection of matrixes with multiple occupancy sites for activating ions is often employed to broaden the spectral range, thereby generating a wide range of spectral emissions. Moreover, co-doping with other luminescent ions can be used to complement the spectral emission. Quantum efficiency and thermal stability are crucial factors in the applications of phosphors. Here, the improvement of phosphor quantum efficiency through flux strategies and the mechanism of defect engineering to enhance thermal stability are highlighted.

Despite extensive research on regulating the luminescence properties of phosphors used for plant growth, there remains many numerous unresolved issues. (1) For light-conversion films, the compatibility between inorganic luminescent materials and resin is poor, which significantly impacts the uniformity and light transmittance of the film. (2) Although the co-doping strategy can broaden the luminescence spectrum of phosphors, it often results in a reduction in luminescence efficiency due to the energy transfer between luminescent ions. (3) The formation of defects in the matrix lattice can enhance the thermal stability of phosphors by trapping electrons, but it can also lead to the intrinsic luminescence reduction in phosphors. (4) The rigid structure of phosphors results in weak thermal vibrations, which are beneficial for reducing non-radiative transitions, but results in a narrow-band emission.

## Figures and Tables

**Figure 1 nanomaterials-13-01715-f001:**
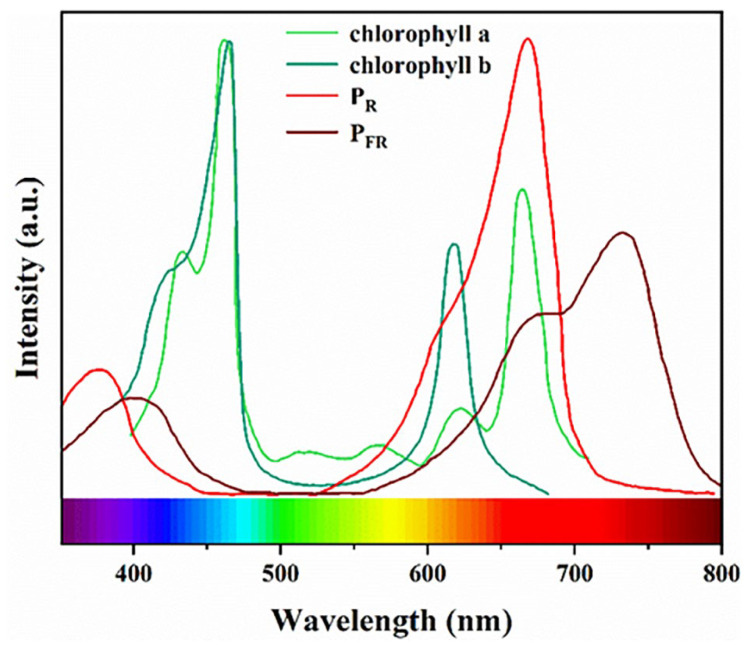
Absorption spectra of plant pigments, including: chlorophyll a, chlorophyll b, photosensitive pigment P_R_, and photosensitive pigment P_FR_.

**Figure 2 nanomaterials-13-01715-f002:**
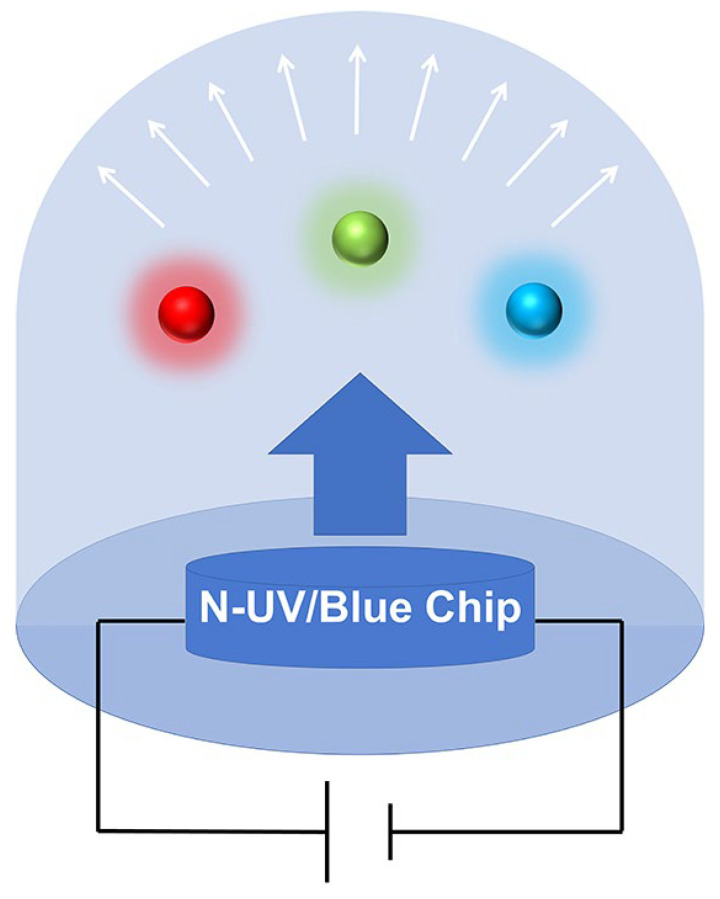
Schematic diagram of the pc-LED structure.

**Figure 3 nanomaterials-13-01715-f003:**
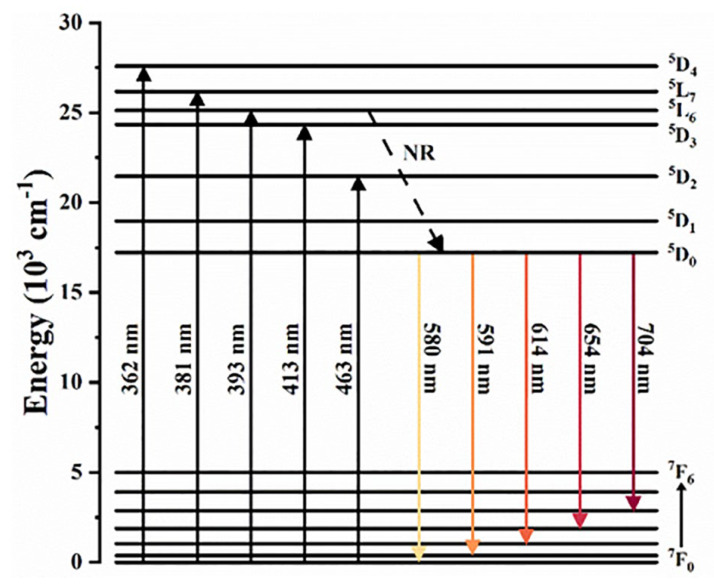
Schematic diagram of the energy level of Eu^3+^.

**Figure 4 nanomaterials-13-01715-f004:**
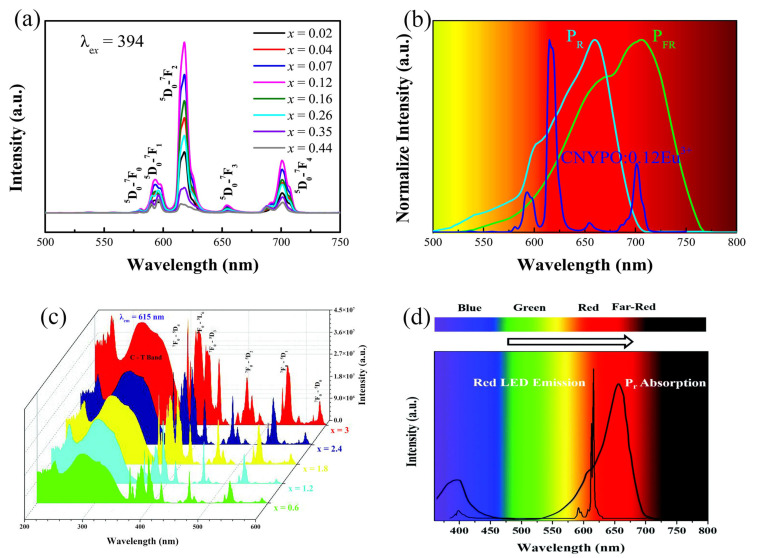
(**a**) Photoluminescence (PL) spectra for Ca_9_LiY_0.667_(PO_4_)_7_: xEu^3+^. (**b**) Comparison of PL spectrum of CNYPO:0.12Eu^3+^ and the absorption spectra of phytochrome P_R_ and P_FR_. (Reprinted from ref. [43]. Copyright of Elsevier, 2020). (**c**) Photoluminescent excitation (PLE) (under λ_em_ = 615 nm) of Li_3_BaSrLa_3−x_Eu_x_(MoO_4_)_8_ where x = 0–3 for selected compositions. (**d**) The spectral overlap between the red LED and phytochrome (P_R_) absorption. (Reprinted from ref. [44]. Copyright of Royal Society of Chemistry, 2021).

**Figure 5 nanomaterials-13-01715-f005:**
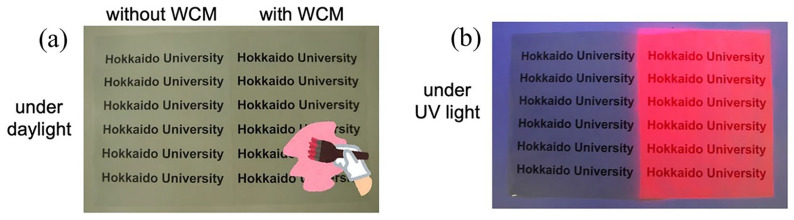
Photographs of agriculture films painted with or without wavelength-converting materials under daylight (**a**) and ultraviolet irradiation (**b**). (Reprinted from ref. [46]. Copyright of Springer Nature, 2022).

**Figure 6 nanomaterials-13-01715-f006:**
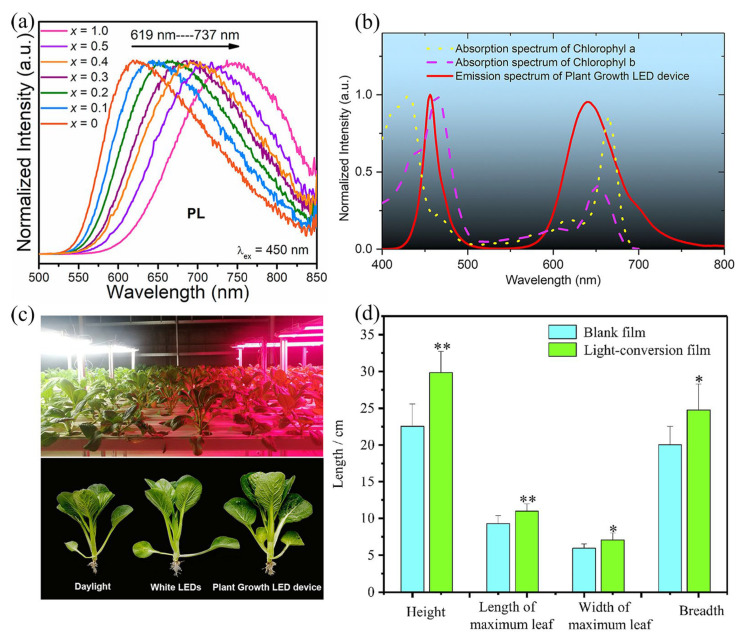
(**a**) Normalized PL spectra of as-prepared (Rb_x_K_1−x_)_3_LuSi_2_O_7_:0.01Eu^2+^ (0 ≤ x ≤ 1). (Reprinted from ref. [51]. Copyright of American Chemical Society, 2020). (**b**) Absorption spectra of Chlorophyll (A and B) and the emission spectrum of Li_2_Ca_2_Mg_2_Si_2_N_6_:1.0%Eu^2+^-based plant-growth LED device. (**c**) Indoor pak-choi cultivation irradiated and digital photographs of pak-choi cultivation irradiated by different lighting sources. (Reprinted from ref. [52]. Copyright of John Wiley and Sons, 2019). (**d**) Biomass of Chinese flowering cabbages obtained under the two films. (Reprinted from ref. [53]. Copyright of Elsevier, 2020).

**Figure 7 nanomaterials-13-01715-f007:**
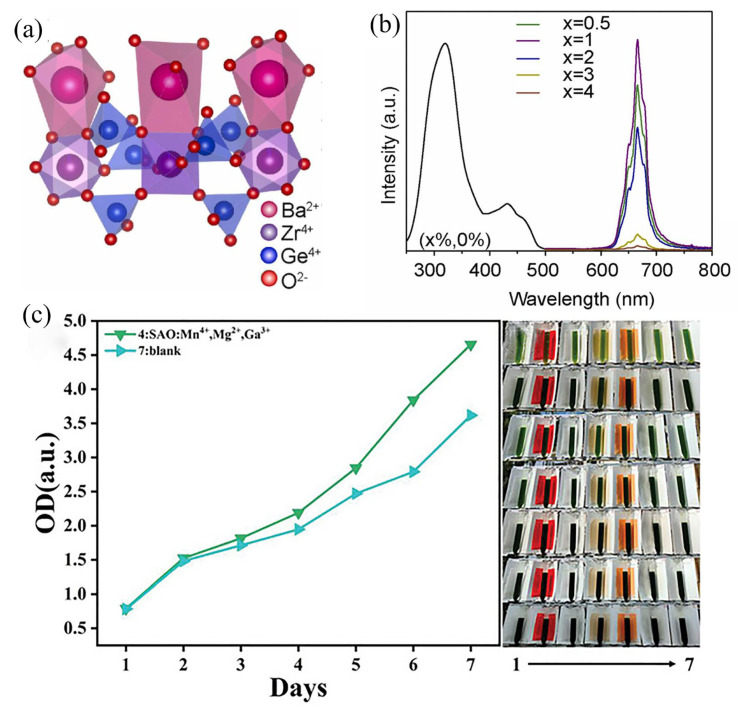
(**a**) Crystal structure of BZGO. (**b**) PL spectra upon excitation by 312 nm and PLE spectra of BZGO:xMn^4+^ (x = 0.5–4%) when monitoring the emission at 666 nm. (Reprinted from ref. [22]. Copyright of Elsevier, 2023). (**c**) Change of optical density of chlorella in tube in a week and photograph of chlorella tube and nearby film (Sr_4_Al_14_O_25_: Mn, Mg, Ga film and blank). (Reprinted from ref. [21]. Copyright of John Wiley and Sons, 2022).

**Figure 9 nanomaterials-13-01715-f009:**
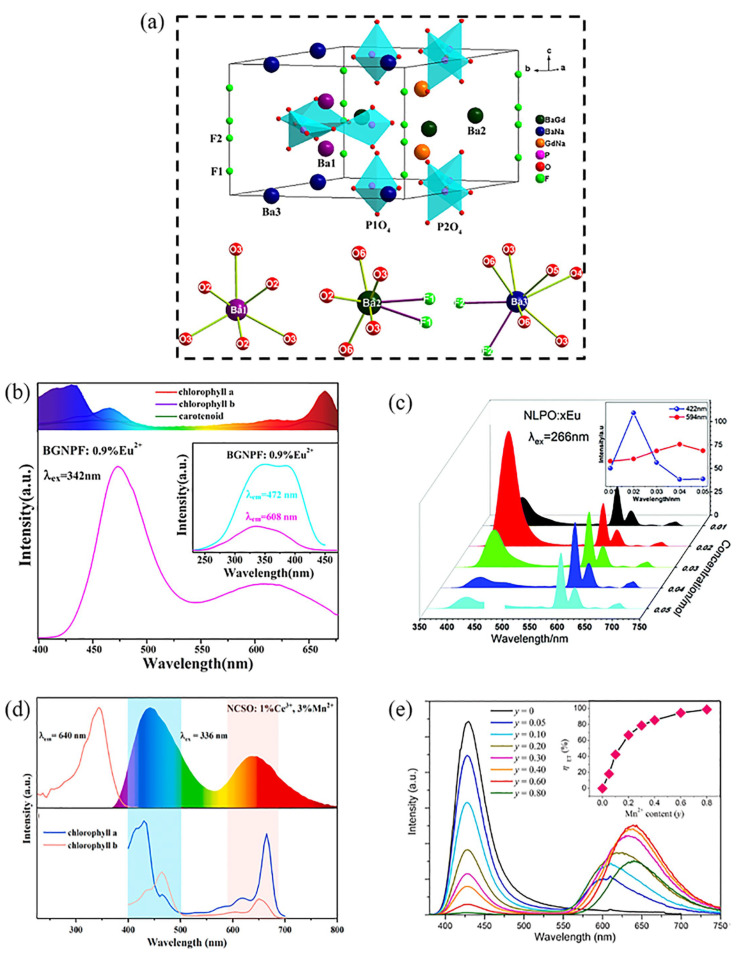
(**a**) Crystal structure of Ba_3_GdNa(PO_4_)_3_F and three kinds of Ba^2+^ ions with a different coordination environment. (**b**) PL and PLE spectra of the phosphor BGNPF:0.9%Eu^2+^ along with the absorption spectra of chlorophyll A, B and carotenoids. (Reprinted from ref. [3]. Copyright of American Chemical Society, 2016). (**c**) Emission (λ_ex_ = 266 nm) spectra of the Na_3_La_2_(PO_4_)_3_:xEu phosphors, the inset shows the peak changes at 422 nm and 594 nm. (Reprinted from ref. [80]. Copyright of Royal Society of Chemistry, 2019). (**d**) Normalized PL and PLE spectra of NCSO:0.01Ce^3+^, Mn^2+^ and absorption spectra of chlorophyll A and chlorophyll B. (Reprinted from ref. [81]. Copyright of Elsevier, 2021). (**e**) The emission spectra of Na_2-x_Mg_2-y_Si_6_O_15_:xEu^2+^, yMn^2+^ (x = 0.02, y = 0–0.80) upon 365 nm excitation. (Reprinted from ref. [82]. Copyright of Elsevier, 2021).

**Figure 11 nanomaterials-13-01715-f011:**
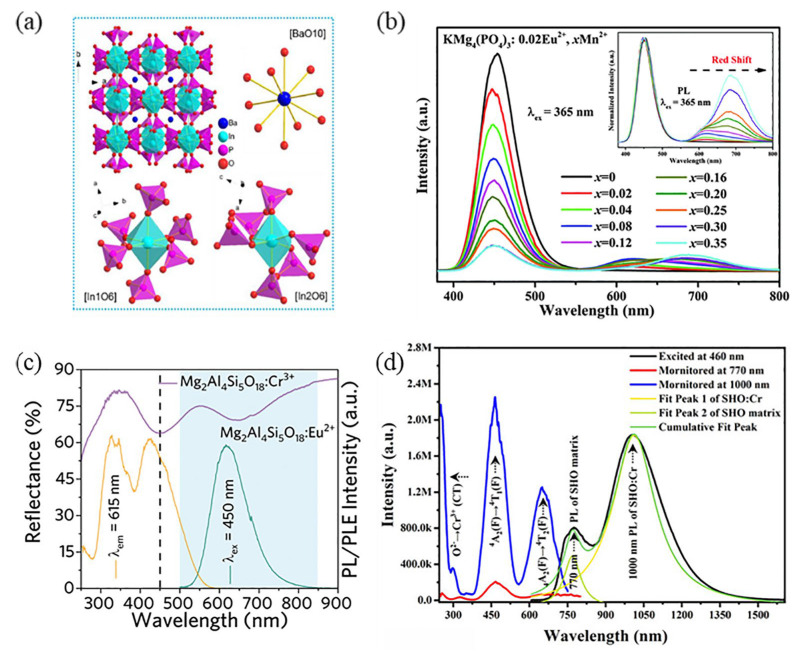
(**a**) Crystal structure of the BIP sample. (Reprinted from ref. [95]. Copyright of Elsevier, 2023). (**b**) The emission spectra of KMg_4_(PO_4_)_3_:0.02Eu^2+^, xMn^2+^ (x = 0–0.35) phosphors. (inset) The normalized emission spectra for KMg_4_(PO_4_)_3_:0.02Eu^2+^, xMn^2+^ (x = 0–0.35) phosphors. (Reprinted from ref. [97]. Copyright of Royal Society of Chemistry, 2022). (**c**) DR spectrum of Mg_2_Al_4_Si_5_O_18_:0.01Cr^3+^, PL and PLE spectra of Mg_2_Al_4_Si_5_O_18_: Eu^2+^. (Reprinted from ref. [98]. Copyright of John Wiley and Sons, 2022). (**d**) Emission and excitation spectra of SHO:0.005Cr. (Reprinted from ref. [99]. Copyright of John Wiley and Sons, 2023).

**Figure 12 nanomaterials-13-01715-f012:**
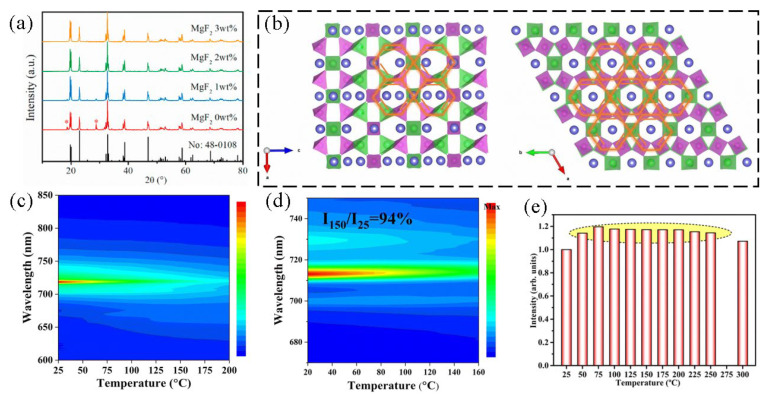
(**a**) XRD patterns of Ca_2_MgWO_6_:1%Bi^3+^ samples prepared with x MgF_2_ (x = 0, 1, 2 and 3 wt%) flux and standard card (JCPDS No. 48-0108) for Ca_2_MgWO_6_. (Reprinted from ref. [102]. Copyright of Elsevier, 2023). (**b**) Crystal structure of LiGaGeO4 and LiAlSiO4. (**c**,**d**) Temperature-dependent PL spectra of LiGaGeO_4_:0.002Cr^3+^ and LiAlSiO_4_:0.02Cr^3+^. (Reprinted from ref. [104]. Copyright of Royal Society of Chemistry, 2022). (**e**) The relationship between luminescence intensity and temperature. (Reprinted from ref. [105]. Copyright of Royal Society of Chemistry, 2022).

**Table 1 nanomaterials-13-01715-t001:** Excitation and emission wavelengths of phosphors for LEDs on indoor plant cultivation.

Phosphor	λ_ex_/nm	λ_em_/nm	Emission Color	Refs.
Ca_3_Al_2_Ge_3_O_12_: Eu^3+^	393	707	Red	[5]
KBa_2_(PO_3_)_5_: Mn^4+^	438	660	Red	[26]
Gd_3_Al_4_GaO_12_: Cr^3+^	360	734	Red	[27]
Sr_2_ScSbO_6_: Mn^4+^	310	700	Red	[28]
SrLa_2_Al_2_O_7_: Mn^4+^	365	731	Red	[29]
Ca_2_BO_3_Cl: Bi^3+^	486	732	Red	[30]
Li_2_ZnTi_3_O_8_: Cr^3+^	360/468	735	Red	[31]
BaLaLiWO_6_: Mn^4+^	332	708	Red	[32]
Li_3_BaSrY_3_(WO_4_)_8_: Eu^3+^	394	615	Red	[33]
Ca_0.995_Mg_2_(SO_4_)_3_: Eu^2+^	554	635	Red	[34]
Ca_1.1_Sr_0.9_SiO_4_: Ce^3+^	365	425	Blue	[4]
La_3_SbO_7_: Bi^3+^	315	520	Blue	[35]
Na_2_BaSr(PO_4_)_2_: Eu^2+^	325	428	Blue	[36]

## Data Availability

Not applicable.

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
