# Peer review of "Recent Advances in Light-Conversion Phosphors for Plant Growth and Strategies for the Modulation of Photoluminescence Properties"

_nanomaterials, 2023, doi:10.3390/nano13111715_

Round 1

Reviewer 1 Report

This review presents the latest development of phosphors for plant growth, with an extensive introduction describing the effects of light on plant growth and the techniques for promoting plant growth. The most used luminescence centers for blue, red and far-red emission are described with their principal photophysical properties. The latest strategies for regulating the spectral position and bandwidth of phosphors emission, for improving their quantum efficiency and thermal stability are discussed at the end of the review.

I find the overall quality of the review acceptable since it gathers a quite large number of phosphors employed for light conversion providing the essential information on their properties in view of this specific application. The introduction is very detailed and somehow redundant but it can be helpful for researchers working in different fields. Paragraph 3.1 and 3.2 seem to be written for a National project as they refer only to China, please revise by enlarging the view for an international journal.

Some terms used in the description of phosphors photophysics are incorrect and misleading, therefore they must be corrected (see below).

Examples of phosphors used in agricultural films as light converter for blue-UV LED are listed for indoor plant cultivation, however the problem of their dispersion in plastic films, the type of polymer used and their degradation issues in greenhouses applications are of interest and should  be mentioned and discussed in the review.

 To my opinion this work can be published on Nanomaterial only after a careful revision of English language.

an example is reported here:

page 2: lines 57-59 “After being photoexcited, they transition from the ground state to the excited state. Since the unstable electrons in the excited state will transition from the excited state to the ground state, there will be a radiation transition” Please note that “transition” is not a verb, “unstable electrons” means probably that the excited state is metastable, “radiation transition” is radiative transition. Similar problems are encountered in lines 200-201, page 5. Line 315, pag.9: line 355 pag11, line 379;

 Other minor points:

Figure 9 is too crowded and difficult to read

The work should be revised by someone expert in scientific English language, to revise some mitakes as “transition” is not a verb, “unstable electrons” means probably that the excited state is metastable, “radiation transition” is radiative transition. Similar problems are encountered in lines 200-201, page 5. Line 315, pag.9: line 355 pag11, line 379

Author Response

Response to Reviewer 1 Comments

The authors thank the Reviewer for the comments and suggestions on our manuscript provided. Each point has been considered carefully and revised the manuscript accordingly (marked in red color). The followings are the detailed responses to each of the comments.

Point 1: Paragraph 3.1 and 3.2 seem to be written for a National project as they refer only to China, please revise by enlarging the view for an international journal.

Response 1: As suggested, we have made changes to the content in lines 148–150, page 4, lines 189–192, page 5, and lines 205-211, page 5. Not to analyze the current state of development of light conversion agricultural films from a domestic perspective.

Point 2: Some terms used in the description of phosphors photophysics are incorrect and misleading, therefore they must be corrected (see below).

an example is reported here:

page 2: lines 57-59 “After being photoexcited, they transition from the ground state to the excited state. Since the unstable electrons in the excited state will transition from the excited state to the ground state, there will be a radiation transition” Please note that “transition” is not a verb, “unstable electrons” means probably that the excited state is metastable, “radiation transition” is radiative transition. Similar problems are encountered in lines 200-201, page 5. Line 315, pag.9: line 355 pag11, line 379;

Response 2: As suggested, we have made a series of modifications to the errors you mentioned. In addition, we have reviewed the entire text and made detailed modifications to the incorrect terminology expressions.

Point 3: Examples of phosphors used in agricultural films as light converter for blue-UV LED are listed for indoor plant cultivation, however the problem of their dispersion in plastic films, the type of polymer used and their degradation issues in greenhouses applications are of interest and should  be mentioned and discussed in the review.

Response 3: As suggested, the key problem in the application of rare earth light conversion agent in light conversion agricultural film is its dispersibility in polymer. Surface modification of light conversion agent is needed to increase its compatibility with polymer. At present, almost all of the reported papers prepare light conversion agricultural film by directly adding light conversion agent into polymer, and the application effect is not ideal.

We have also done some tentative work in the surface modification of light conversion agent, by mainly two strategies.One is to use the coating method to cover the surface of light conversion agent with a layer of polymer shell, and then mix it into the polymer; The other is to use silane coupling agent, ethylene wax, stearic acid and other modifiers to modify the light conversion agent and then add into the polymer, so that it is evenly dispersed. Both of these methods can be used to distribute the light conversion agent evenly in the polymer, however, more further work needs to do in the near future.

Point 4: To my opinion this work can be published on Nanomaterial only after a careful revision of English language.

Response 4: As suggested, we have carefully polished the English writing.

Point 5: Figure 9 is too crowded and difficult to read.

Response 5: As suggested, we have made changes to Figure 9, as shown below:

Reviewer 2 Report

I read the paper titled "Recent advances in the field of light conversion phosphors for plant growth and Strategies for the Modulation of the photoluminescence properties"

I found the review interesting because it provides a wide collection of cases describing materials and approaches developed in the last years for the production of LED for plant growth.

The paper is well-written and the topic is focused and well-realized by accurately describing the technique and results obtained right now in this field.

The manuscript is potentially interesting for readers of Nanomaterials. As a reviewer, I still have some comments and suggestions.

1. I suggest improving the discussion about the effect of phosphors concentration in the composite to tailor the resulting luminescence by also considering these studies

 Influence of the Ce: YAG amount on structure and optical properties of Ce:YAG-PMMA composites for white LED Zeitschrift fur Physikalische ChemieOpen AccessVolume 230, Issue 9, Pages 1219 - 1231September 2016

Concentration-dependent multi-color humic acid-based carbon dots for luminescent polymer composite films. J Mater Sci 57, 1069–1083 (2022). https://doi.org/10.1007/s10853-021-06606-6

(2005). Influences of matrices and concentrations on luminescent characteristics of Eu(TTA)3(H2O)2/polymer composites. Journal of Luminescence, 114(3-4), 187-196. https://doi.org/10.1016/j.jlumin.2005.01.002

2.       line 739 "are summarized" correct the size

3.       Please standardize Figure and Fig. in the text 

Author Response

Response to Reviewer 2 Comments

The authors thank the Reviewer for the comments and suggestions on our manuscript provided. Each point has been considered carefully and revised the manuscript accordingly (marked in red color). The followings are the detailed responses to each of the comments.

Point 1: I suggest improving the discussion about the effect of phosphors concentration in the composite to tailor the resulting luminescence by also considering these studies

1.Influence of the Ce: YAG amount on structure and optical properties of Ce:YAG-PMMA composites for white LED Zeitschrift fur Physikalische ChemieOpen AccessVolume 230, Issue 9, Pages 1219 - 1231September 2016

2.Concentration-dependent multi-color humic acid-based carbon dots for luminescent polymer composite films. J Mater Sci 57, 1069–1083 (2022). https://doi.org/10.1007/s10853-021-06606-6(2005).

3.Influences of matrices and concentrations on luminescent characteristics of Eu(TTA)3(H2O)2/polymer composites. Journal of Luminescence, 114(3-4), 187-196. https://doi.org/10.1016/j.jlumin.2005.01.002

Response 1: As suggested, we have carefully read the papers recommended and other papers with similar research. This aspect is set out in lines 622-644 of the manuscript.

Point 2: line 739 "are summarized" correct the size

Response 2: Done as suggested.

Point 3: Please standardize Figure and Fig. in the text

Response 3: Done as suggested.
